# Tolerance of Three Quinoa Cultivars (*Chenopodium quinoa* Willd.) to Salinity and Alkalinity Stress During Germination Stage

**Vasile Stoleru** [1,*] , **Cristina Slabu** [2] , **Maricel Vitanescu** [1] , **Catalina Peres** [1] , **Alexandru Cojocaru** [1] , **Mihaela Covasa** [2] **and Gabriela Mihalache** [1,3,*]

[1] Department of Horticultural Technologies, University of Agricultural Sciences and Veterinary Medicine "Ion Ionescu de la Brad", 3 M. Sadoveanu, 700440 Iasi, Romania; vitanescu_maricel@yahoo.com (M.V.); catalina_peres@yahoo.com (C.P.); cojocaru.alexandru@yahoo.com (A.C.)

[2] Department of Plant Physiology, University of Agricultural Sciences and Veterinary Medicine "Ion Ionescu de la Brad", 3 M. Sadoveanu, 700440 Iasi, Romania; cristinaslabu@yahoo.com (C.S.); miha_bologa@yahoo.com (M.C.)

[3] Integrated Center of Environmental Science Studies in the North East Region (CERNESIM), The "Alexandru Ioan Cuza" University of Iasi, 11 Bd. Carol, 700506 Iasi, Romania

* Correspondence: vstoleru@uaiasi.ro (V.S.); gabriela.mihalache.gm@gmail.com (G.M.)

**Abstract:** Salinity and alkalinity are two of the main causes for productivity losses in agriculture. Quinoa represents a better alternative for global food products such as rice and wheat flour due to its high nutritional value and abiotic stress tolerance. Three cultivars of quinoa seeds (*Titicaca*, *Puno* and *Vikinga*) originating from Denmark were used in the experiments. The seeds were germinated under the action of three different salts ($NaCl$, $Na_2SO_4$, $Na_2CO_3$) at 0–300 mM for five days and the germination rate was calculated. Biometric measurements (radicle and hypocotyls lengths) andbiochemical determinations (proline) were performed in order to quantify the tolerance and the effects of salt and alkali stresses on the three quinoa cultivars. The germination rates showed that all cultivars were affected by the presence of salts, especially at 300 mM. The most sensitive cultivar to salts was *Titicaca* cultivar which evinced the lowest germination rate, regardless of the salt and the concentration used. On the other hand, *Puno* and *Vikinga* cultivars showed the best tolerance to the saline and alkaline stresses. Among the salts used, $Na_2CO_3$ had the most detrimental effects on the germination of quinoa seeds inhibiting the germination by ~50% starting with 50 mM. More affected was the growth of hypocotyls in the presence of this salt, being completely inhibited for the seeds of the *Puno* and *Titicaca* cultivars. *Vikinga* cultivar was the only one able to grow hypocotyls at 50 and 100 mM $Na_2CO_3$. Also, this cultivar had a high adaptability to $NaCl$ stress when significant differences were observed for the germination rates at 200 and 300 mM as compared to 0 mM $NaCl$, due to the proline production whose content was significantly greater than that of the untreated seeds. In conclusion, the tolerance of the three quinoa cultivars to saline and alkali stress varied with the salt type, salt concentration and tested cultivar, with the *Vikinga* and *Puno* cultivars showing the best potential for growing under saline conditions.

**Keywords:** *Chenopodium quinoa*; salt tolerance; germination; proline

## 1. Introduction

Soil salinity is one of the major problems humankind faces in the twenty first century. Inappropriate irrigation practices, use of salt-rich irrigation water and poor drainage conditions are the main causes for an increase in the salt content of soils [1,2]. It is estimated that of the total cultivated lands (1.5 Bha), 23%

(340 Mha) are salt-affected (saline), while 37% (560 Mha) are sodium-affected (sodic) [3]. The presence of high concentrations of salts and sodium in soil affects its fertility and limits the plant productivity and distribution [4–8]. When saline soils also contain $CO_3^{2-}$ the detrimental effects on plants are more pronounced. Germination is the first process highly affected by salinity and alkalinity because of ion toxicity and the negative effects on osmosis. Salinity and alkalinity not only delay seed germination but may, depending on plant tolerance, inhibit it completely. Therefore, finding salt-alkaline-tolerant species and cultivars is the key to successful germination and seedling establishment [9].

A crop that received a great attention in the recent years from both scientists and farmers is quinoa (*Chemopodium quinoa* Willd.). The UN Food and Agriculture Organization (FAO) selected quinoa as the plant species that can assure food security in this century due to its nutritional features and also for its important tolerance against several abiotic stresses including salinity [10]. Quinoa can be cultivated for its edible seeds and also for the leaves [11]. Studies have shown that the seeds are gluten-free and contain a wide range of minerals, vitamins, antioxidants, and proteins whose quality is comparable to that of casein, as well as high amounts of essential amino acids such as lysine that is commonly found in cereals only in a limited quantity [12,13]. In Denmark, quinoa was selected as a protein crop for organic feed and was also recommended for people with coeliac disease. Due to its extraordinary nutritional traits, the global market of quinoa is currently increasing rapidly [14]. At the moment, in Europe there are nine registered cultivars, five originating from Netherland (Carmen, Atlas, Pasto, Riobamba and Red Carina), three from Denmark (*Titicaca*, *Puno* and *Vikinga*) and one from France (Jessie) [14]. Quinoa is also known for its high salt tolerance, which is greater than that of other cereals as barley, wheat, corn or vegetable crops such as spinach, onion, carrots or asparagus [15]. Many studies regarding the tolerance of quinoa to salt have been focused on *Titicaca* cultivar and on the effect of NaCl on the germination of seeds [12,16–23], but little is known about the other cultivars and the effect of other salts such as $Na_2SO_4$ or $Na_2CO_3$.

In this study the tolerance of the three cultivars originating from Denmark (*Titicaca*, *Puno* and *Vikinga*) to salinity (NaCl, $Na_2SO_4$) and alkalinity ($Na_2CO_3$), with a particular emphasis on seed germination, were investigated and compared. As far as we know, no information about the germination of *Vikinga* cultivar under salinity or alkalinity stress can be found. Regarding the tolerance of *Puno* cultivar to salinity and alkalinity, the existing data are scarce and are focused on other aspects than germination. Knowing the differences in quinoa's cultivar tolerance to chloride and sulfate salinity can bring important information for a more precise use of this crop in lands affected by soil salinity.

## 2. Materials and Methods

### 2.1. Plant Material

Three quinoa cultivars originating from Denmark 'Titicaca' (T), 'Puno' (P) and 'Vikinga' (V) were used in experiments. *Titicaca* and *Puno* were recorded in Europe in 2009 and *Vikinga* in 2015 [14]. Prior to be used in experiments, seeds were stored at 5–10 °C.

### 2.2. Germination Assay

Germination of the seeds was evaluated in petri dishes of 9 cm in diameter, on filter paper. A number of 50 seeds per dish were placed on filter paper moistened with 5 mL of distilled water or NaCl, $Na_2SO_4$, $Na_2CO_3$ solutions at four different concentrations: 50 mM, 100 mM, 200 mM and 300 mM. The filter paper was moistened only once, just at the beginning of the experiment. Germination was carried out in a growth chamber at a temperature of 25 °C in the dark with a relative humidity of 70%, for five days [12]. The experiment was performed in triplicate. Seeds were considered germinated when the radicle had extended at least 1 mm. The seed germination was recorded daily for five days.

The germination rate was calculating using the following formula: germination rate% = (number of germinated seeds/number of total seeds) × 100 [9].

## 2.3. Biometric Measurements

The length of the radicle and hypocotyls were measured using a slide rule as follows: the radicles were measured after two days after germination and at the end of the experiment (five days) and the hypocotyls were measured at day five.

## 2.4. Proline Determination

Proline was determined at the end of the experiment on whole the plant material by using the modified method of Bates, 1973: 0.5 g of plant material was homogenized in 5 mL of 3% aqueous sulfosalicylic acid and the homogenate centrifuged for 10 minutes at 5000 rpm. Then, 2 mL of the supernatant was reacted with 2 mL ninhydrin acid and 2 mL of glacial acetic acid in a test tube for 1 h at 100 °C. The reaction was terminated in an ice bath. The reaction mixture was extracted with 4 mL toluene and mixed vigorously for 15–20 s. The colored toluene aliquot was aspirated from the aqueous phase and the absorbance was read at spectrophotometer at 520 nm, by using toluene as blank [24].

The content of proline was calculated as nmol.mg$^{-1}$ FW by using the formula: ($Abs_{extract}$ − blank)/Slope × $Vol_{extract}$/$Vol_{aliquot}$ × 1/FW, where: $Abs_{extract}$ = the absorbance of the extract, Blank (expressed as absorbance) and Slope = expressed as absorbance in nmol$^{-1}$ are determined by linear regression, $Vol_{extract}$ = the total volume of the extract, $Vol_{aliquot}$ = the volume used in the assay and F.W = the amount of plant material (mg) [25].

## 2.5. Statistical Analysis

The statistical evaluation of the data was carried out by one-way and two-way ANOVA with replication using salt type, cultivars and salt concentration as factors. Tukey tests were performed in order to estimate the significant difference between the treatment means, and among the cultivars. Differences between groups were considered statistically significant when $p < 0.05$.

## 3. Results and Discussion

### 3.1. Salinity and Alkalinity Effect on the Germination of the Seeds

Three cultivars of quinoa (*Titicaca*, *Puno*, *Vikinga*) originating from Denmark were chosen to be studied for their ability to germinate under the stress of two salts known for causing salinity in soils (NaCl and $Na_2SO_4$) and one salt responsible for alkalinity—$Na_2CO_3$. Most of the studies are focusing on the tolerance of quinoa, especially of *Titicaca* cultivar, to NaCl [6,10,12,13,26], one of the main factors for soil salinity [27], but little is known about the ability of quinoa seeds to germinate under $Na_2SO_4$ or $Na_2CO_3$ stresses [28]. Knowing the differences in quinoa's tolerance to chloride and sulfate salinity can bring important information for a more precise use of this crop in lands affected by soil salinity. Regarding the effects of alkalinity on quinoa's germination no information exists at the moment.

The concentrations of the salts used in the experiments (50, 100, 200 and 300 mM) correspond to three classes of soil salinity: slightly saline (50 mM or 5 dS m$^{-1}$), moderately saline (100 mM or 9 dS m$^{-1}$) and strongly saline (200 and 300 mM, respectively 18 and 27 dS m$^{-1}$) [29].

The data showed that the germination of *Titicaca* cultivar reached its highest rate at day four, when 73% of the untreated (0 mM) seeds were germinated. This rate remained the same until the end of the experiment. All the treatments affected the quinoa seed of *Titicaca* cultivar germination regardless of the applied concentration.

The NaCl treatment, after 24 h, significantly affected the seeds germination at all the concentrations used in the experiment compared to the untreated seeds. Therefore at 50, 100 and 200 mM NaCl, the germination rate at day five was at least 25% less than that of the seeds with 0 mM NaCl, while at 300 mM concentration the difference in germination was larger, 59%. Our data are in disagreement with those obtained by Panuccio et al. (2014) who observed that NaCl did not have significant effects on the germination percentage of *Titicaca* quinoa cultivar seeds when used in concentrations of 100, 200, 300 or 400 Mm [12]. In respect to the differences in germination rate between the four concentrations of

NaCl used in experiment, in the first three days, no significant differences were observed between them (50, 100, 200 and 300 mM), however, in the fourth and fifth days significant differences in the germination rate were noticed for the seeds treated with 300 mM NaCl, compared with 50, 100 and 200 mM. Also, for the same concentration, the germination rate remained the same starting with the third day (14%) (Figure 1).

Regarding the effects of $Na_2SO_4$ stress on *Titicaca* cultivar, there are studies showing that for many plants there is a trend of greater sulfate tolerance than chloride tolerance [30]. Such tolerance was also found in four Chilean lowland quinoa cultivars [28], and our results showed that the germination rate at 100 and 200 mM was 10%, respectively 25% less compared with the germination rate at the same concentrations of NaCl. Also, as in the case of NaCl, the germination of *Titicaca* cultivar seeds in the presence of $Na_2SO_4$, regardless of the concentration used, was significantly affected compared to the untreated seeds. Significant differences were seen also between 200 mM or 300 mM and 0, 50 or 100 mM concentration. The same germination rate was for 50 and 100 mM treatment and also for 200 and 300 mM (Figure 1).

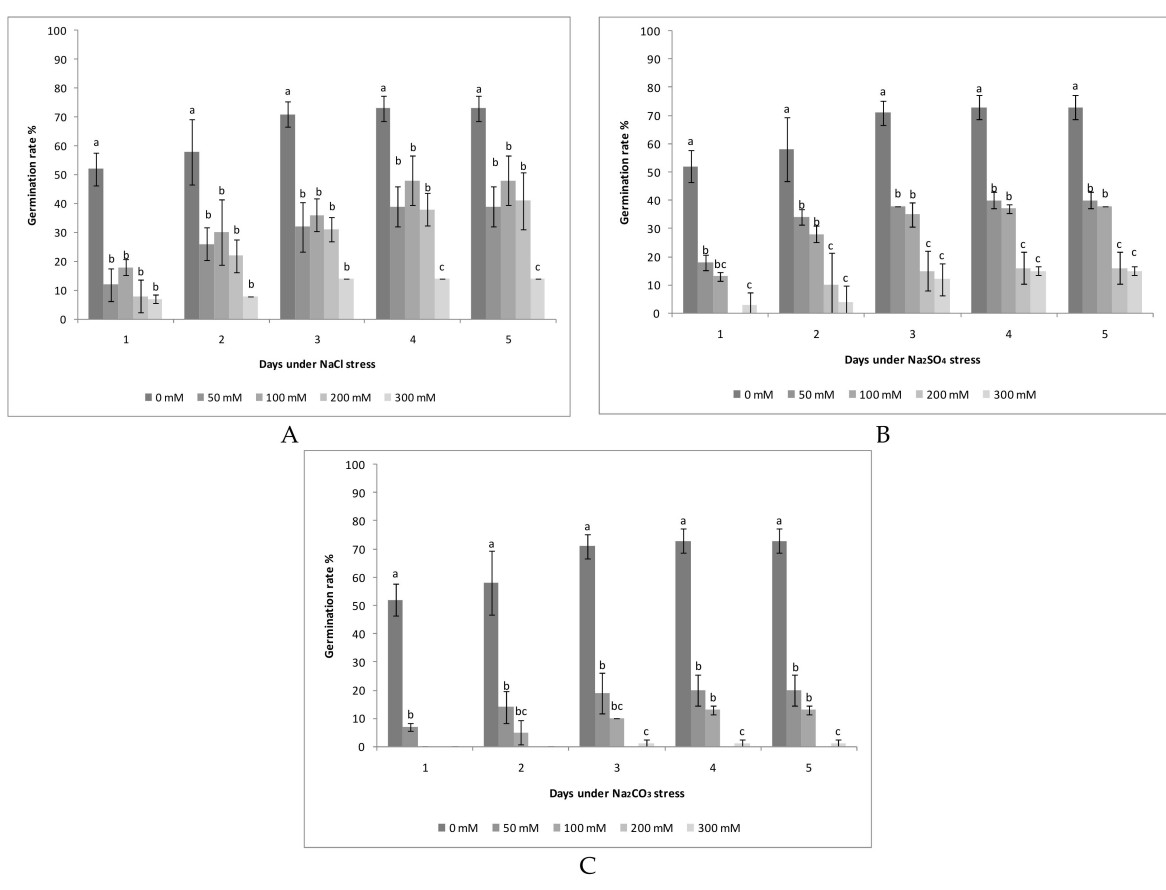

**Figure 1.** Seed germination rate of *Titicaca* cultivar under NaCl (**A**), $Na_2SO_4$ (**B**) and $Na_2CO_3$ (**C**) stresses. Different letter means significant differences, according to Tukey test. Error bars indicate mean ± SD.

The data of the effect of $Na_2CO_3$ treatment on *Titicaca* cultivar showed that the germination of quinoa seeds was inhibited starting with 200 and 300 mM. Only at 50 and 100 mM $Na_2CO_3$, 20%, respectively 13% of the seeds were able to germinate until the end of the experiment. Significant differences were seen between the untreated and treated seeds (Figure 1). Also, a delay in the germination process of the seeds treated with 100 $Na_2CO_3$ mM was observed, the emergence of the radicle being visible after two days of incubation.

*Puno* cultivar is less studied than *Titicaca* cultivar. Information about its tolerance to salinity is scarce and is focused on other aspects than germination. For example, Adolf et al. (2012) included

it in their studies when investigated the varietal differences of quinoa's tolerance to NaCl in a pot experiment [16]. Further, Shabala et al. (2013) used it in an experiment regarding the mechanisms involved in salt tolerance by comparing cultivars of different geographical origin [31].

In our experiments focused on germination, quinoa seeds of *Puno* cultivar which was not under salinity or alkalinity stress (0 mM) germinated in a proportion of 87% after three days of incubation (Figure 2). Seeds germination rate was not affected by NaCl stress until 300 mM concentration of the salt, when no significant differences were seen between 0 mM and 50, 100 or 200 mM. Only at 300 mM NaCl the germination was 21% less than that of the untreated plants (0 mM) (Figure 2). Similar data were obtained by Gonzalez and Prado (1992), on *Sajama* cultivar when the germination rate was not affected at 100 and 200 mM as compared with 0 mM [32]. Further, Ruiz-Carroso et al. (2011), registered for one (BO78) of the four Chilean quinoa genotypes tested in the presence of 0, 150 and 300 mM, a germination inhibition at 300 mM [6].

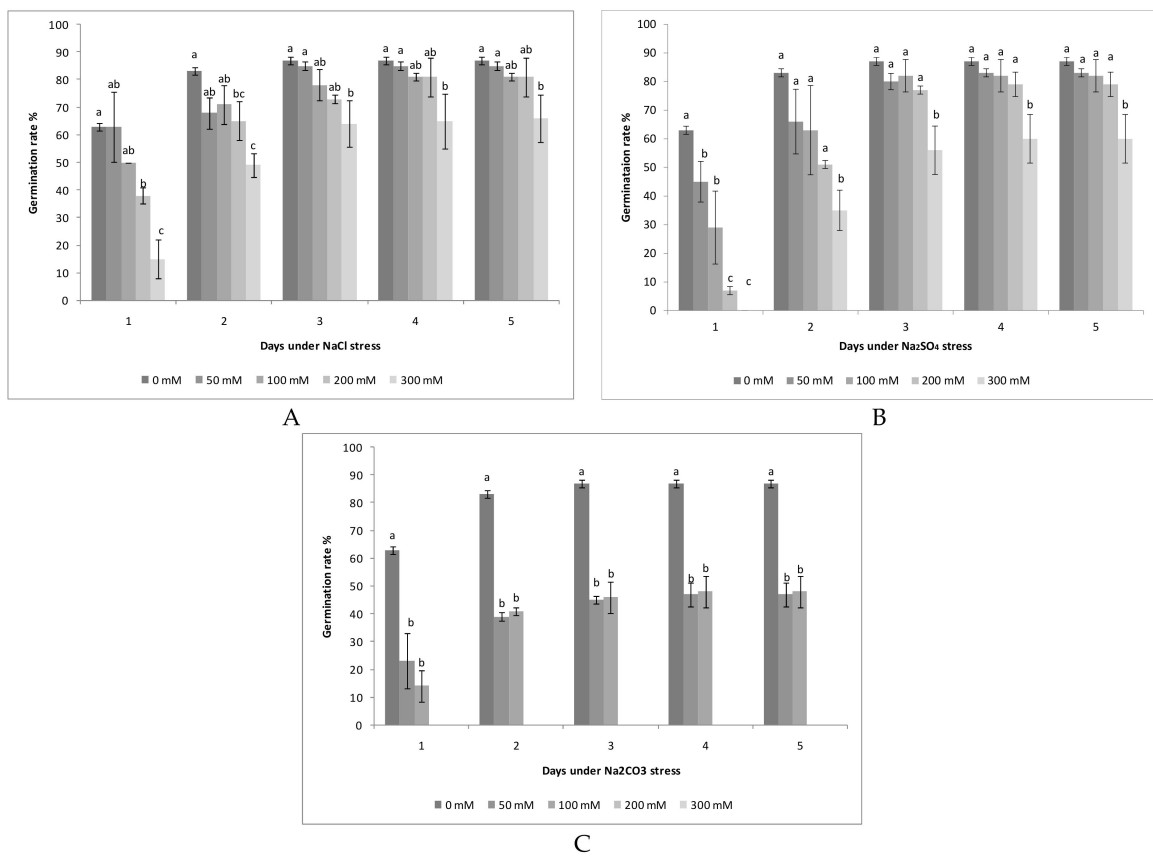

**Figure 2.** Seed germination rate of *Puno* cultivar under NaCl (**A**), $Na_2SO_4$ (**B**) and $Na_2CO_3$ (**C**) stresses. Different letter means significant differences, according to Tukey test. Error bars indicate mean ± SD.

Regarding the $Na_2SO_4$ stress, as in the case of NaCl treatment, the germination rate was significantly less only at 300 mM compared to the untreated seeds (Figure 2). Seeds germination was not affected by $Na_2SO_4$ when it was used in concentrations of 50, 100 or 200 mM, no differences being registered compared with 0mM.

As in the case of quinoa seeds cultivar *Titicaca*, *Puno* cultivar germination was highly affected by the presence of $Na_2CO_3$ stress. The seeds were able to germinate only at 50 mM (47%) and 100 mM (48%), germination being completely inhibited at 200 and 300 mM concentrations (Figure 2).

As far as we know, no information about the germination of *Vikinga* cultivar under salinity or alkalinity stress can be found. Our data showed that the germination rate of *Vikinga* cultivar untreated seeds after four days of incubation reached its maximum value—78%. The NaCl treatment did not affect the germination regardless of the concentration used in the experiment. Therefore, no difference

between the treated and untreated seeds was seen at the end of the five days of incubation (Figure 3). Only a delay in germination at 200 and 300 mM was seen in the first two days. Similar results were obtained by Ruffino et al. (2010), on *Sajama* cultivar at 250 mM NaCl and by Prado et al. (2010) at 400 mM NaCl on the same cultivar [33,34].

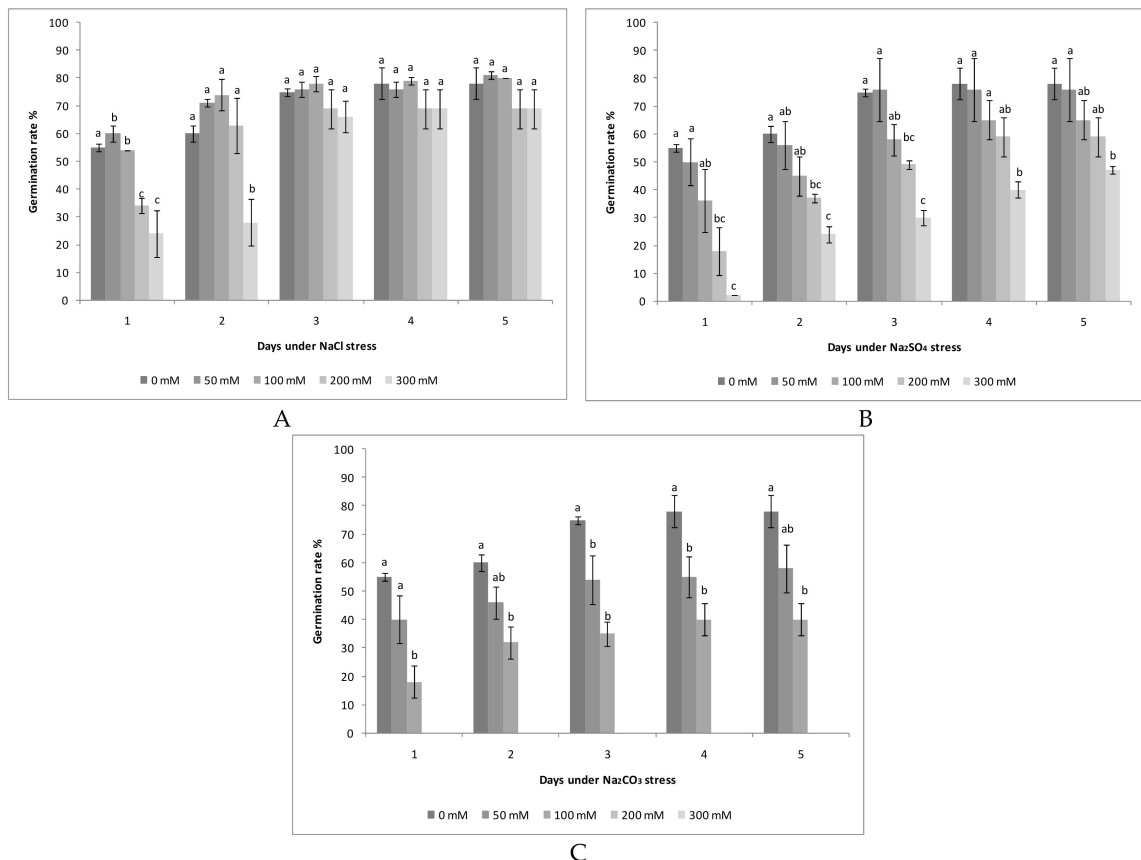

**Figure 3.** Seed germination rate of *Vikinga* cultivar under NaCl (**A**), $Na_2SO_4$ (**B**) and $Na_2CO_3$ (**C**) stresses. Different letter means significant differences, according to Tukey test. Error bars indicate mean ± SD.

The $Na_2SO_4$ treatment affected the seed germination only at 300 mM, when only 47% succeeded to germinate and significant differences between the untreated and treated seeds were seen.

The alkalinity stress induced by $Na_2CO_3$ inhibited the quinoa seeds germination at concentrations of 200 and 300 mM. At 50 mM $Na_2CO_3$, 58% of the seeds germinated and at 100 mM, only 40% germinated. Significant differences between 50 and 100 mM concentrations compared to 0 mM were registered.

### 3.2. Tolerance of Quinoa Cultivars to the Salt and Alkali Stress

The tolerance and the adaptability of a plant to stress conditions such as salinity or alkalinity are tested by the production of different organic compounds such as proline [35,36]. Studies, have shown that quinoa can tolerate moderate (150 mM NaCl) to high levels (750 mM NaCl) of salinity [15]. Moreover, the ability of germination and seedling establishment under saline conditions of quinoa plants is dependent on the cultivar [37,38]. In our experiments we compared the tolerance and the ability of three Denmark quinoa cultivars to adapt to salt and alkali stress taking into account the germination rates and we correlated it with the proline production.

Our data showed that the three cultivars of quinoa used in the experiment had different behaviour depending on the salt type to which they were exposed. Moreover, without the presence of any stress, differences in the germination rate were observed between the cultivars (Figure 4A). Therefore, after

five days of incubation, the highest germination rate was registered for *Puno* cultivar seeds (87%), followed by *Vikinga* cultivar (78%) and *Titicaca* cultivar (73%). At the end of the experiment, significant differences were observed only between *Puno* and *Titicaca* cultivar.

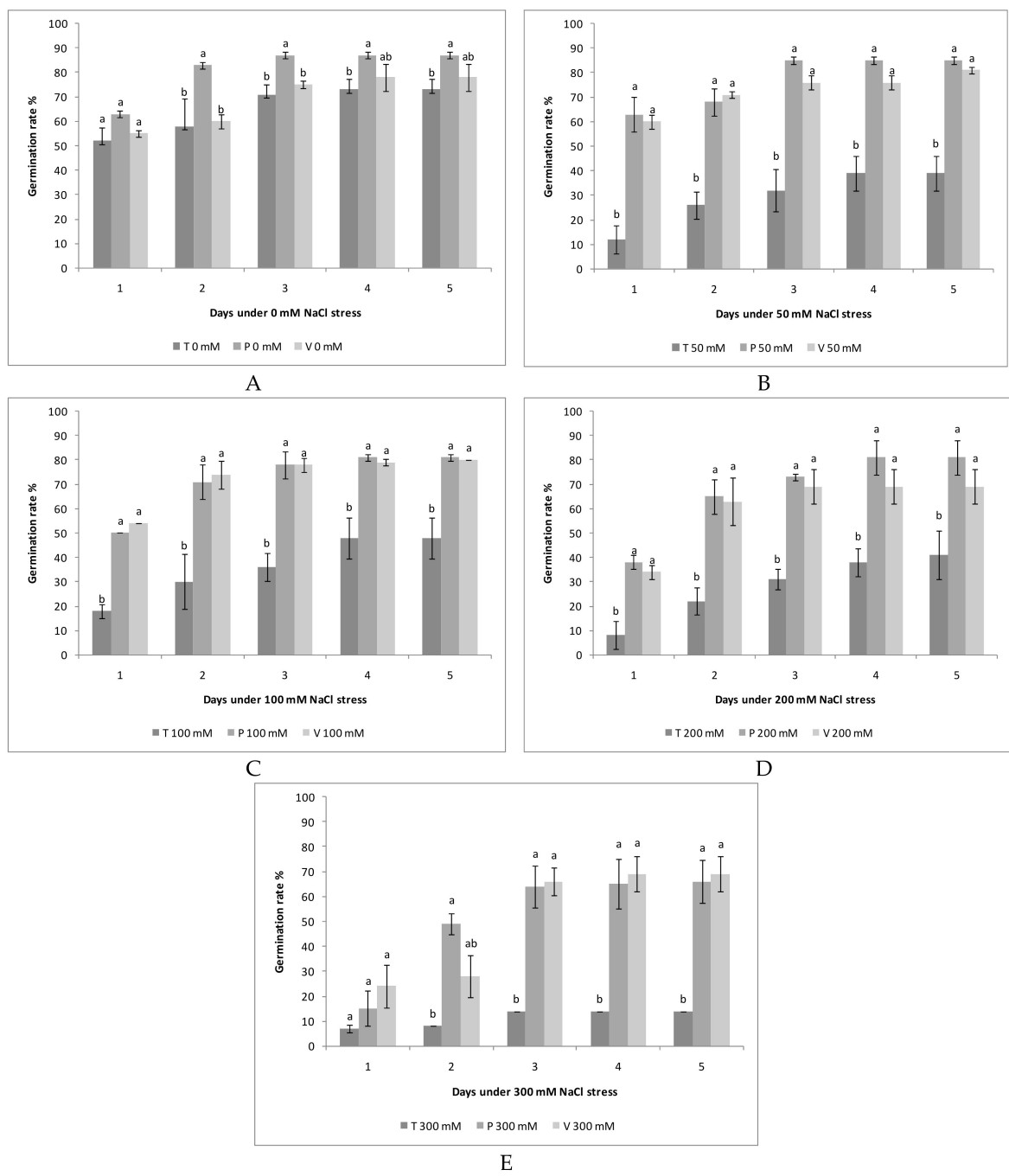

**Figure 4.** Comparisons between *Titicaca* (T), *Puno* (P) and *Vikinga* (V) cultivars under 0 mM NaCl (**A**), 50 mM NaCl (**B**), 100 mM NaCl (**C**), 200 mM NaCl (**D**), 300 mM NaCl (**E**) stresses. Different letter means significant differences, according to Tukey test. Error bars indicate mean ± SD.

When NaCl stress under different concentrations (50, 100, 200, 300 mM) was present, the germination of *Titicaca* cultivar seeds were the most affected regardless of the concentration used. The germination rate of *Titicaca* seeds cultivar was almost 50% less than that of the seeds of *Puno* and *Vikinga* cultivars at 50, 100, 200 and 300 mM NaCl (Figure 4B–E). According to Shabala et al. (2013), *Titicaca* and *Puno* cultivars based on the geographic origin are adapted to a low-salinity habitat, their

salinity stress tolerance being determined by it. Even though our data are in agreement with those made by Shabala et al. (2013) when it comes to *Titicaca* cultivar, for *Puno* cultivar a significant higher germination was registered compared with that of *Titicaca* regardless the NaCl concentration used. Therefore, based on our data, *Titicaca* cultivar is more sensitive to NaCl stress than *Puno* or *Vikinga* cultivars. In parallel, the analysis of proline production by the seeds of *Titicaca* cultivar, demonstrated no significant differences between the different NaCl concentrations (Table 1).

However, a high adaptability on the NaCl stress was register for *Vikinga* cultivar, when significant differences were observed for proline at 200 and 300 mM compared to 0 mM NaCl (Table 1). For the same concentrations, the proline production was significantly higher than that of the untreated plants. Due to proline production, *Vikinga* seed rate germination was almost the same as that of the seeds of Puno cultivar (Figure 4). Thus, we can assume that due to proline production, *Vikinga* cultivar cultivar was able to adapt to high concentrations of NaCl as compared to *Titicaca*.

**Table 1.** Proline content of *Titicaca*, *Puno* and *Vikinga* cultivars under NaCl, $Na_2SO_4$ and $Na_2CO_3$ stress after 5 days of germination. Different letters in the same column mean significant differences between treatments, according to Tukey test ($p < 0.05$).

| Salt | Concentration (mM) | Proline Content (nmol $mg^{-1}$ FW) | | |
|---|---|---|---|---|
| | | *Titicaca* | *Puno* | *Vikinga* |
| **NaCl** | 0 | 0.59 [a] | 0.81 [a] | 0.47 [c] |
| | 50 | 0.38 [a] | 0.8 [a] | 0.56 [bc] |
| | 100 | 0.96 [a] | 1.14 [a] | 0.77 [abc] |
| | 200 | 0.97 [a] | 1.14 [a] | 1.02 [ab] |
| | 300 | 0.97 [a] | 1.06 [a] | 1.19 [a] |
| **$Na_2SO_4$** | 0 | 0.59 [a] | 0.81 [a] | 0.47 [a] |
| | 50 | 0.61 [a] | 0.46 [a] | 0.99 [a] |
| | 100 | 1.11 [a] | 0.43 [a] | 0.91 [a] |
| | 200 | 0.63 [a] | 0.55 [a] | 0.54 [a] |
| | 300 | 0.91 [a] | 1.01 [a] | 0.81 [a] |
| **$Na_2CO_3$** | 0 | 0.59 [a] | 0.81 [a] | 0.47 [a] |
| | 50 | 0.81 [a] | 0.49 [a] | 0.38 [a] |
| | 100 | 0.47 [a] | 0.57 [a] | 0.55 [a] |
| | 200 | 0.48 [a] | 0.74 [a] | 0.56 [a] |
| | 300 | 0.51 [a] | 0.98 [a] | 0.50 [a] |

Regarding the stress produced by $Na_2SO_4$, at 50 mM, seeds of *Vikinga* and *Puno* cultivar had the best germination rate (76%, also 83% starting with day 3, until the end of the experiment), significant differences being registered as compared with *Titicaca* cultivars (Figure 5A). Also, the proline content of *Vikinga* seeds was almost twice compared with that of the control (0 mM), which did not happen for *Puno* or *Titicaca* cultivar (Table 1). Therefore, we presume that the better germination rate of *Vikinga* cultivar exposed to 50 mM $Na_2SO_4$ can be attributed to proline production.

When the concentration of the salt was increased to 100 mM, the best germination rate was registered for *Puno* cultivar (82%) followed by *Vikinga* cultivar (65%) (Figure 5C). The differences between the cultivars were not significant. However, it was interesting to observe that the germination of the seeds of *Puno* cultivar was almost the same at 100 and 200mM $Na_2SO_4$ (82%, respectively 79%), as compared with 50 mM (83%), probably due to either a stimulatory effect that higher concentrations of $Na_2SO_4$ can have on this cultivar, taking into account that no significant differences were seen in the proline content between the concentrations used in the experiment or due to the high resistance of it to this salt. Stimulatory effects of sulfates are not unusual. Orlovskyet al. (2016) in a study regarding another specie from *Chenopodiaceae* family, *Salicornia europaea* L., saw that at concentrations between 0.5%–3%, sulfates can have a stimulatory effect on germination [39]. At 300 mM $Na_2SO_4$, the germination rate of all the cultivars was less than in the other concentrations, with *Puno* and

*Vikinga* giving the best values. Also, it was observed that if for *Titicaca* and *Vikinga* cultivars the sulfate treatment reduced the germination rate compared to NaCl treatment, quinoa seeds of *Puno* cultivar was not affected by it, the germination rate registered being almost the same for both the salt treatments. This can suggest that *Puno* cultivar has a natural resistance to these salts.

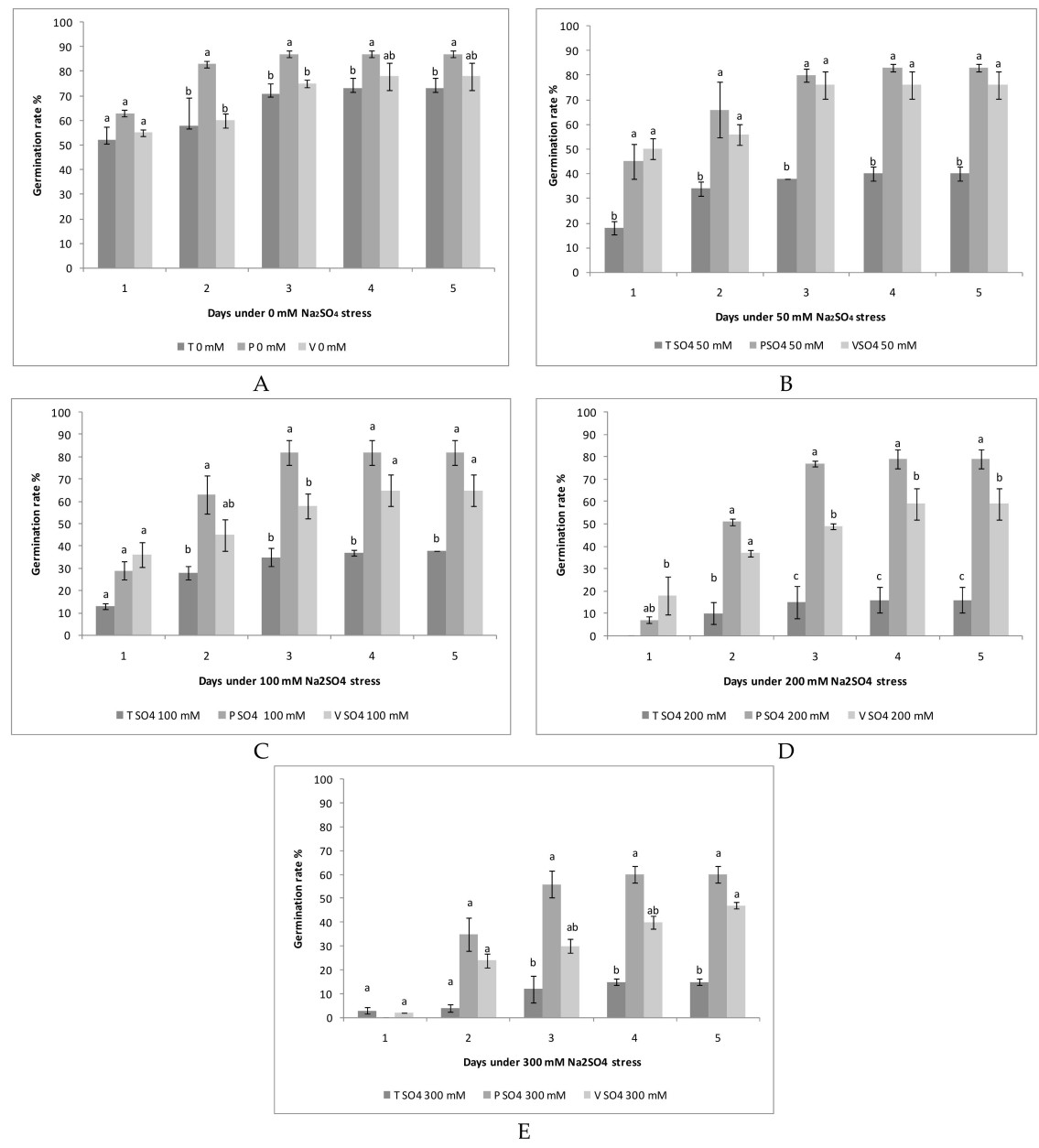

**Figure 5.** Comparisons between *Titicaca* (T), *Puno* (P) and *Vikinga* (V) cultivars under 0 mM $Na_2SO_4$ (**A**), 50 mM $Na_2SO_4$ (**B**), 100 mM $Na_2SO_4$ (**C**), 200 mM $Na_2SO_4$ (**D**), 300 mM $Na_2SO_4$ (**E**) stresses. Different letter means significant differences, according to Tukey test. Error bars indicate mean ± SD.

For *Titicaca* cultivar, as in the case of NaCl, the $Na_2SO_4$ highly affected the germination rate of the seeds especially at 200 and 300 mM. This cultivar registered the lowest germination rate, regardless of the concentration used in the experiment. Moreover, no significant differences were seen between the proline content of the seeds exposed to $Na_2SO_4$.

Finally, the exposure of quinoa seeds cultivar to $Na_2CO_3$ resulted to have the most detrimental effect on the germination rate. The seeds of all the cultivars were able to germinate only at 50 and 100 mM, at 200 and 300 mM the germination being completely inhibited (Figure 6). At 50 mM, 58% of

*Vikinga* seeds cultivar germinated, followed by *Puno* cultivar with 47% and *Titicaca* 20%. Significant differences were registered between *Puno* or *Vikinga* and *Titicaca* germination rate. No differences were observed between the proline content of the seeds regardless of the concentrations used and the cultivar (Table 1).

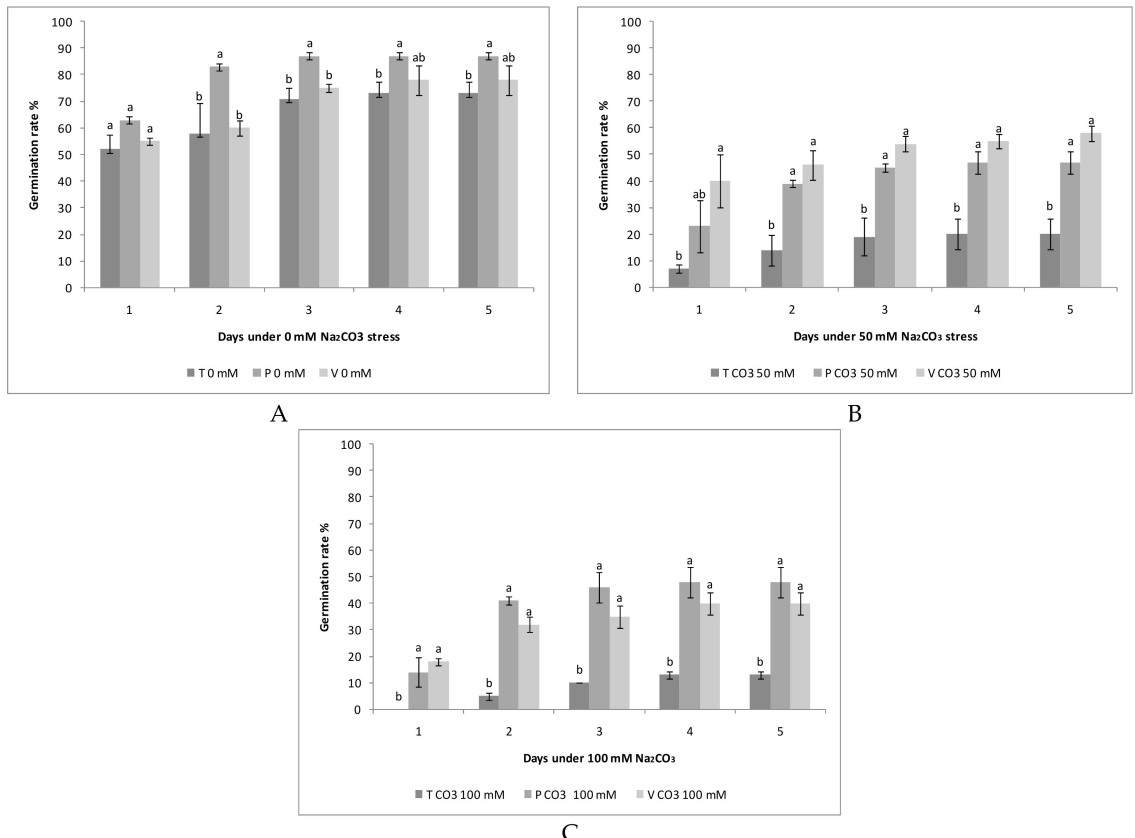

**Figure 6.** Comparisons between *Titicaca* (T), *Puno* (P) and *Vikinga* (V) cultivars under 0 mM Na$_2$CO$_3$ (**A**), 50 mM Na$_2$CO$_3$ (**B**), 100 mM Na$_2$CO$_3$ (**C**) stresses. Different letter means significant differences, according to Tukey test. Error bars indicate mean ± SD.

Usually, the salt induced inhibition of seed germination can be attributed to osmotic stress as a result of the decreasing rate of water uptake, or to ion toxicity as a consequence of the excessive intake of ions such as Na$^+$ which may change certain enzymatic or hormonal activities inside the seed [12,26,40,41]. Instead, alkaline stress, even if it has the same ion (Na$^+$) as saline stress factor, the added influence of high pH stress enhance the osmotic and ionic toxicity on seeds, but also can destroy the embryo [9]. This may be the reason of the lack of germination of the seeds exposed at 200 and 300 mM Na$_2$CO$_3$ regardless of the cultivar used.

*3.3. Effects of the Salinity and Alkalinity on the Growth of Radicles and Hypocotyls*

The growth of radicle and hypocotyls were affected by the saline and alkali stress for most of the concentrations used in the experiment and for all the cultivars, significant differences being registered between the control seeds (0 mM) and treatments (Table 2).

Under NaCl the best values for the radicle length were registered for the *Vikinga* cultivar. Also, for the same cultivar, at 300 mM NaCl, the seeds were able to grow the hypocotyls as compared to *Titicaca* and *Puno* cultivars which were able to develop radicals, but not to form hypocotyls (Table 2).

**Table 2.** Effect of NaCl, $Na_2SO_4$ and $Na_2CO_3$ stresses on the growth of radicle and hypocotyls of *Titicaca*, *Puno* and *Vikinga* cultivars after five days of germination. Different letters in the same column mean significant differences between treatments, according to Tukey test ($p < 0.05$).

| Treatment | | *Titicaca* | | *Puno* | | *Vikinga* | |
|---|---|---|---|---|---|---|---|
| Salt | Concentration (mM) | Radicle Length (mm) | Hypocotyls Length (mm) | Radicle Length (mm) | Hypocotyls Length (mm) | Radicle Length (mm) | Hypocotyls Length (mm) |
| NaCl | 0 | 8.1 [ab] | 23.3 [a] | 12.8 [a] | 27.2 [a] | 20.5 [a] | 31.8 [a] |
| | 50 | 7.2 [ab] | 18.1 [b] | 7.4 [b] | 20.2 [b] | 12.6 [bc] | 22.8 [ab] |
| | 100 | 10.4 [a] | 9.5 [c] | 7.4 [b] | 16.8 [bc] | 18.5 [ab] | 17.1 [b] |
| | 200 | 6.1 [ab] | 2.7 [d] | 5.5 [bc] | 10.4 [c] | 11.8 [c] | 10.7 [bc] |
| | 300 | 4.9 [b] | 0 [d] | 4.8 [c] | 0 [d] | 6.8 [c] | 5.8 [c] |
| $Na_2SO_4$ | 0 | 8.1 [a] | 23.3 [a] | 12.8 [a] | 27.2 [a] | 20.5 [a] | 31.75 [a] |
| | 50 | 6.1 [a] | 11.6 [b] | 7.8 [b] | 15.8 [b] | 15 [ab] | 9.9 [bc] |
| | 100 | 5.2 [ab] | 8.5 [bc] | 5.9 [bc] | 9.45 [b] | 8.2 [bc] | 11.8 [b] |
| | 200 | 2.9 [bc] | 0 [c] | 4.6 [bc] | 0 [c] | 5.1 [c] | 2.4 [cd] |
| | 300 | 1.6 [c] | 0 [c] | 3.5 [c] | 0 [c] | 2.6 [c] | 0 [d] |
| $Na_2CO_3$ | 0 | 8.1 [a] | 23.3 [a] | 12.8 [a] | 27.2 [a] | 20.5 [a] | 31.8 [a] |
| | 50 | 2.2 [b] | 0 [b] | 3.1 [b] | 0 [b] | 8.4 [b] | 8.9 [b] |
| | 100 | 1.5 [b] | 0 [b] | 2 [b] | 0 [b] | 2.7 [c] | 3.9 [c] |
| | 200 | 0 [c] | 0 [b] | 0 [c] | 0 [b] | 0 [c] | 0 [d] |
| | 300 | 0 [c] | 0 [b] | 0 [c] | 0 [b] | 0 [c] | 0 [d] |

Previous studies have demonstrated that if the seeds manage to sprout and take root, then there is a strong possibility that it will develop to the next stage [34]. Our data showed that irrespective of the radicle formation for *Titicaca* and *Puno* cultivars, the hypocotyls formation was inhibited. The same situation was registered for $Na_2SO_4$ at 200 mM for *Titicaca* and *Puno* seeds, also at 300 mM for all the three cultivars used in the experiments and for $Na_2CO_3$, at 50 and 100 mM for *Titicaca* and *Puno* cultivars and at 200 or 300 mM for all the cultivars. The ability of *Vikinga* seeds to grow the hypocotyls at 300 mM NaCl was probably due to the proline content, which was produced in a significant amount compared to the untreated plants (Table 1). For *Titicaca* and *Puno* cultivars, no differences in the proline content was registered regardless of the salt and concentration used in the experiments. Even though no significant differences were seen in the proline content for *Vikinga* cultivar under $Na_2CO_3$, the seeds were able to grow hypocotyls at 50 and 100 mM as compared to *Titicaca* and *Puno* seeds, which could not. This might have happened because of the seed dimensions of *Vikinga* cultivar, which are bigger (2 mm) than the seeds (~1.5 mm) of the other two cultivars. Due to their larger size, probably a bigger dry matter and energy accumulated inside the embryos. Demir and Mavi (2008), demonstrated in a study regarding the effect of salt and osmotic stresses on the germination of peppers seeds that the larger the reserve material accumulated, the greater the tolerance of the seeds to stress conditions [42]. Another reason for this might be the structure of the seed coat, that can protect the embryos until there is a specific concentration of the stress factor, in our case, 200 mM $Na_2CO_3$ [43].

## 4. Conclusions

The tolerance of the three quinoa cultivars to saline and alkali stresses varied with the salt type, salt concentration and tested cultivar. The cultivars which showed the best potential for germinating and growing under saline conditions were *Vikinga* and *Puno* cultivars. On the other hand, the most sensitive cultivar to salts was *Titicaca* cultivar which evinced the lowest germination rate, regardless the salt and the concentration used in the experiment. The germination rates demonstrated that all the cultivars were affected by the presence of salts, especially at 300 mM regardless of the salt used. Among the salts, $Na_2CO_3$ had the most detrimental effects on the germination of quinoa seeds inhibiting the germination by ~50% starting with 50 mM. More affected was the growth of hypocotyls in the presence of this salt, which was completely inhibited for the seeds belonging to *Puno* and *Titicaca* cultivars. *Vikinga* cultivar was the only one able to grow hypocotyls at 50 and 100 mM $Na_2CO_3$. Further, this

cultivar had a high adaptability to NaCl stress due to proline production, the content of which was significantly higher than that of the untreated seeds.

**Author Contributions:** V.S., G.M. and M.V. conceived the research idea and experimental protocol, coordinated the research and wrote the manuscript; G.M. and V.S. critically commented on the manuscript draft; C.P. and A.C. were involved in performed experiment; C.S., and M.C. manage in proline analyses.

**Funding:** This research did not receive external funding.

**Acknowledgments:** The authors wish to thank Jitareanu Carmen and Modiga Beatrice for their helpful assistance of proline analyses.

**Conflicts of Interest:** The authors declare no conflict of interest.

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
