# Peer review of "Tolerance of Three Quinoa Cultivars (Chenopodium quinoa Willd.) to Salinity and Alkalinity Stress During Germination Stage"

_agronomy, doi:10.3390/agronomy9060287_

Reviewer 1 Report

An original study highlighting the effects of three quinoa cultivars to salinity and alkalinity stress. The science is sound but there is a need for the manuscript to be copy edited by a Native English speaker. I have inserted some edits in the attached manuscript with suggestions to improve it. 

Author Response

RESPONSE TO REVIEWER 1 (GREEN)

Comment 1: An original study highlighting the effects of three quinoa cultivars to salinity and alkalinity stress. The science is sound but there is a need for the manuscript to be copy edited by a Native English speaker. I have inserted some edits in the attached manuscript with suggestions to improve it. 

Response 1: As it has been suggested, the following edits have been done:

Line 20: “also” has been replaced with “also”

Line 27: “in a percentage of” has been replaced with “by”

Line 29: “belonging to” has been replaced with “of the”

Line 30: “demonstrated to have” has been replaced with “had”

Line 32: “significant bigger” has been replaced with “significantly greater”

Line 34: “with the” was added

Line 39: “one century” has been replaced with “first Century”

Line 44: “contain also” has been replaced with “also contain”

Line 58: the topic of the sentence was changed as follows: “selected as a protein crop for organic feed andwas alsorecommended for people with coeliac”

Line 81: “kept” has been replaced with “stored”

Line 94: the topic of the sentence was changed as follows: “The length of the radicle and hypocotyls were measured using a slide rule as follows”

Line 112: “results” has been replaced with “data”

Line 130: “result” has been replaced with “data”

Line 135: “as” has been deletedfrom “as compared”

Line 137:  “bigger” has been replaced with “larger”, also “results”with “data”

Line 140: “got” has been deleted from “rate got”

Line 143: “was” has been replaced with “were”

Line 144: “as” has been deleted from “as compared”

Line 146: “which are saying” has been replaced with “showing”

Line 149: “as” has been deleted from “as compared”

Line 151: “as” has been deleted from “as compared”

Line 153: the topic of following sentence “The same rate germination was get for 50 and 100 mM treatment, also for 200 and 300 mM” was changed as “The same germination rate was for 50 and 100 mM treatment and also for 200 and 300 mM”

Line 158: “results” has been replaced with “data”, also “shows” with “showed”

Line 174: “was” has been replaced with “were”

Line 176: “results” has been replaced with “data”

Line 181: “significant” has been replaced with “significantly”, also “as” has been deleted from “as compared”

Line 183: “as” has been deleted from “as compared”

Line 191: “results” has been replaced with “data”

Line 196: “or” has been replaced with “and”

Line 198: “of it” has been deleted from 47% “of it”

Line 202: “germinated” has been added, also “as” has been deleted from “as compared”

Line 206: “done” has been replaced with “tested”

Line 212: “results” has been replaced with “data”

Line 213: “depending on the salt to which were exposed” was changed to “depending on the salt type to which they were exposed”, also “behavior” has been replaced with “behaviour”

Line 215: “biggest” has been replaced with “highest”

Line 223: “results” has been replaced with “data”

Line 225: “better” has been replaced with “higher”, also “as” was deleted from “as compared”

Line 226: “results” has been replaced with “data”

Line 234: “as” has been deleted from “as compared”

Line 246: “as” has been deleted from “as compared”, also “thing that” was replaced with “which”

Line 259: “to” has been deleted from “in”

Line 260: “…Puno and Vikingagiving the best values” was changed to “… with Puno and Vikingagiving the best values”

Line 261: “cultivar” has been changed to “cultivars”, also “as” has been deleted from “as compared”

Line 292:  “It was interesting to observe that under NaCl the best values for the radicle length were registered for Vikinga cultivar.” was changed to “Under NaCl the best values for the radicle length were registered for Vikinga cultivar.”

Line 299: “results” has been replaced with “data”

Line 305: For the rest of the cultivars…” was changed to “For Titicaca and Puno cultivars…”, also “as” has been deleted from “as compared”

Line 307: “Anyway” was changed to “Even though”

Line 309: “couldn’t” has been replaced with “could not”, also “seeds” was changed to “seed”

Line 313: “bigger” has been replaced with “larger”

Line 322: “Anyway the germination rates…” was changed to “The germination rates …”

Line 325: “in a percentage of” has been replaced with “by”

Line 328: “demonstrated to have” has been replaced with “had”

Line 329: “significant bigger” has been replaced with “significantly greater”

Reviewer 2 Report

This is a relevant manuscript, but some issues should be addressed first.

In this study three different salts and three cultivars of quinoa were assessed. The authors should report, for each figure and table, the relative ANOVA table showing the main effects and their interactions. Only in this way the results will be fully understood.

The authors should point out in the title and in the manuscript that the results regarding the quinoa seedlings.

Author Response

RESPONSE TO REVIEWER 2 (BLUE)

Comment 1: In this study three different salts and three cultivars of quinoa were assessed. The authors should report, for each figure and table, the relative ANOVA table showing the main effects and their interactions. Only in this way the results will be fully understood.

Response 1: Annex 1 regarding the ANOVA table showing the main effects and their interactions has been added.

Comment 2: The authors should point out in the title and in the manuscript that the results regarding the quinoa seedlings.

Response 2: Our determinations and results focused on the effects of saline and alkali stress on germination of the seeds. To be clearer the title of the manuscript was changed as follows: “Tolerance of Three Quinoa Cultivars (Chenopodium quinoa Willd.) to Salinity and Alkalinity Stress During Germination Stage”

Round  2

Reviewer 1 Report

Page 7, Line 208 - Replace "taking into accounts" with "taking into account".

Author Response

Thanks for your comments. We corrected (L210-211).